# Recovery of Water from Secondary Effluent through Pilot Scale Ultrafiltration Membranes: Implementation at Patras’ Wastewater Treatment Plant

**DOI:** 10.3390/membranes11090663

**Published:** 2021-08-28

**Authors:** Dimitris Zagklis, Fotios K. Katrivesis, Varvara Sygouni, Lamprini Tsarouchi, Konstantina Tsigkou, Michael Kornaros, Christakis A. Paraskeva

**Affiliations:** 1Laboratory of Transport Phenomena and Physicochemical Hydrodynamics (LTPPH), Department of Chemical Engineering, University of Patras, GR26504 Patras, Greece; dimitris_2502@yahoo.gr (D.Z.); sygouni@upatras.gr (V.S.); 2Department of Environmental Engineering, University of Patras, GR30100 Agrinio, Greece; fkatrivesis@gmail.com; 3Institute of Chemical Engineering Sciences, FORTH/ICE-HT, GR26504 Patras, Greece; 4Municipal Water Supply and Sewerage Company of Patras, GR26333 Patras, Greece; lamprinitsarouhi@yahoo.gr; 5Laboratory of Biochemical Engineering and Environmental Technology (LBEET), Department of Chemical Engineering, University of Patras, GR26504 Patras, Greece; ktsigkou@chemeng.upatras.gr (K.T.); kornaros@chemeng.upatras.gr (M.K.)

**Keywords:** membrane filtration, sewage water, ultrafiltration, sewage treatment plant, tertiary treatment, case study, pilot scale

## Abstract

Fresh water shortages affect larger areas each year due to the increased human population combined with climate change. Reuse of treated sewage water (mostly for nonpotable uses) can have a significant impact on reducing water scarcity. Ultrafiltration membranes are widely considered as a very good candidate for the remediation of this type of water. The case of Patras’ sewage treatment plant was examined for the treatment of its secondary settling tank effluent using a pilot ultrafiltration unit to produce permeate water suitable for reuse according to Greek legislation. The physicochemical characteristics of the membrane permeate stream showed significant improvements in the quality of the produced water. Turbidity was reduced by 99%, total suspended solids were decreased by more than 94%, while COD was reduced by 37%. *E. coli* and *Enterococcus* were detected at high concentrations in the feed stream but were eliminated in the membrane permeate. The results presented herein indicate that the installed equipment is capable of producing improved quality water suitable for reuse even with the strictest limits imposed by Greek legislation.

## 1. Introduction

Wastewater produced from several kinds of human activities is an important untapped resource [1,2]. Fresh water shortages affect larger areas each year due to the increased human population combined with climate change [3,4,5]. Although large quantities of municipal wastewater could be recovered and reused, they usually flow into water bodies (mainly to the sea) [6]. Recovery of such water could lead to lower needs for freshwater extraction and partially prevent aquifer depletion.

The renewal of freshwater supplies is carried out through the natural water cycle (evaporation, precipitation, and runoff) and is driven by global climatic forces. The disturbance in these forces caused by human activity and increased water extraction has led two thirds of the human population to inhabit areas affected by water scarcity [7]. In southern Europe, water reuse is usually driven by high water demand in high density population areas (competition between touristic and agricultural demand vs. municipal demand) [8]. The problem of water scarcity is increasingly affecting the Mediterranean region and is only expected to become more intense with the progressive land desertification caused by climate change and human activity [9,10].

Several treatment techniques have been developed for wastewater treatment and recovery [11]. The produced/treated water can be used for different activities in agricultural, industrial, and urban uses [12,13]. Several regulations have been imposed to define the possible uses of recovered water, depending on its source and characteristics. Typically, reclaimed water is used for activities that consume large quantities of water. In the Mediterranean region, 44% of reclaimed water is reused for irrigation and 37% for environmental applications [8].

The reuse of treated sewage water (mainly for nonpotable uses) can have a significant impact in reducing water scarcity especially in places with very limited access to fresh water. Ultrafiltration membranes are widely considered as a suitable method for the reclamation of this type of water, offering excellent permeate characteristics [14,15,16,17].

The reuse of treated sewage wastewater in Greece is regulated by Greek legislation and more specifically by the Joint Ministerial Decision (JMC), 145116_2011, “Establishment of measures, conditions and procedures for reuse of treated wastewater and other provisions” [18], and the Joint Ministerial Decision, JMC 191002_2013, Amendment of JMC 145116_2011 [19]. The possible uses for treated sewage wastewater include irrigation, enrichment of underground aquifers, urban and suburban uses, and reuse in industrial processes. The treated water quality criteria are summarized in Table 1.

According to Greek legislation, the reuse of treated wastewater for irrigation may be distinguished as two types based on the type of crop, the irrigation system, and public access to the irrigated area. Restricted irrigation applies only to crops with products that are consumed after heat or other treatment, or are not intended for human consumption, or do not come into direct contact with soil, such as animal crops, industrial crops, meadows, trees (not including fruit trees), and provided that the crops are not in contact with the soil during harvesting, seed crops. As for the irrigation system, the sprinkling method is not allowed. Public access to the irrigated area is also not allowed. Where access to humans or animals other than users is possible, additional measures should be taken on a case-by-case basis, such as fencing, designation of a restrictive zone for certain uses of the irrigated area, and prohibition of grazing animals for a certain period after irrigation [18,19].

Unrestricted irrigation applies, inter alia, to all other types of crops such as vegetables, vines, or crops with products that are consumed raw, and floriculture. Unrestricted irrigation permits various methods of using recovered water, including sprinkling, and no access restrictions are required [18,19].

To prevent organic matter accumulation in groundwater during underground aquifer enrichment, which may harm future uses of the aquifer’s groundwater, a sufficient degree of treatment for the removal of organic matter is required, in case of direct enrichment by drilling under pressure or by gravity at selected boreholes. Thus, in addition to secondary biological and possibly tertiary treatment, at least ultrafiltration or an alternative advanced treatment method of equivalent efficacy is needed. In the case of enrichment through soil filtration, the soil must be appropriate and of sufficient depth. Additional advanced treatment methods may be avoided if sufficient retention of organics is achieved through soil filtration [18,19].

Reuse of treated wastewater for urban and suburban activities mainly refers to urban and suburban green, forest areas, recreation, natural remediation, firefighting, street cleaning, nondrinking uses, and household activities. Reuse possibilities include mainly watering of concentrated green areas, such as forests, groves, cemeteries, slopes and traffic islands, golf courses, public parks, courtyards, hotel and leisure facilities, water for recreation, for the cleaning of roads and sidewalks, for decorative fountains, for the creation of artificial, or for the maintenance of natural lakes or wetlands, etc. [18,19].

Reuse of wastewater in the industry involves applications such as use for cooling, replenishment of boiler water, and utilization for various industrial processes. This reuse does not apply to industries of products intended for human consumption. For other industrial uses, additional treatment may be required, depending on the type of industrial reuse [18,19].

In this work, the case of Patras’ sewage treatment plant was examined. The quantitative and qualitive characteristics of the secondary settling tank effluent were used to identify the specifications of the required ultrafiltration unit that would produce permeate water with characteristics that would allow its reuse. After the successful installation of the equipment at Patras’ treatment plant for the treatment of the secondary settling tank effluent, the installed membrane unit was tested regarding its efficiency. Qualitive and quantitative data were obtained through a series of experiments, regarding the flux of the membrane permeate, the applied transmembrane pressures, the physicochemical characteristics of the feed, permeate and concentrate streams, as well as the presence of pathogens in these streams.

The aim of this analysis was to ensure the efficiency of the installed equipment in producing permeate water that could be reused according to national legislation. The experiments were carried out across several days, in order to ensure a range of different feed characteristics. The results presented herein can be used to identify the most appropriate reuse application of the reclaimed water, according to legislation.

## 2. Materials and Methods

The installed ultrafiltration unit was a complete, autonomous, compact, and low-pressure filtration system. It included a mild washing subsystem (air–water) and a chemical cleaning system (air–water and chemicals) controlled by automated electric valves. The filtration and cleaning operation was continuous and automatic, via programmable controller (PLC) and display control. The unit included all the necessary sensors and measuring instruments (pressure, flow, turbidity, etc.) for monitoring the three water streams (feed, permeate, and concentrate) as well as any other instruments/components deemed necessary. Due to the automatic and autonomous operation of the unit, minimum supervision was required. The flow diagram of the process is presented in Figure 1 and the technical characteristics of the equipment are presented in Table 2.

The system consisted of eight LITREE LH3-1060 V membrane modules, each supplying 2.5 m^3^/h of permeate. Assuming a daily operation of the system equal to 20 h, maximum capacity can be calculated equal to 400 m^3^/d. With 90% recovery, a permeate yield of 360 m^3^/h can be achieved, and with a more modest prediction of 70% recovery rate, the daily permeate production is equal to 280 m^3^/h.

The unit consisted of two separate membrane arrays (four modules each), to enable up to 50% of the system’s operation while one of the arrays is in cleaning or maintenance mode. The unit employs hollow fiber membrane modules, with a nominal pore size of 0.01 μm, Molecular Weight Cutoff 50,000 Da. Filtration was achieved by liquid pumping at low pressure flow using a constant and low-pressure feed pump (one pump per treatment line), from the outer to the inner part of the hollow fiber membrane, removing particles larger than 0.01 μm. Larger particles remained on the outer surface of the membrane.

To protect the membrane modules, the system inlet was equipped with a stainless-steel prefilter with a pore size of 100–150 μm. The membrane permeate ended up in the chlorination tank. An appropriate volume plastic tank was also installed for the storage needs of the cleaning chemicals.

Removal of the solids on the outer surface of the membranes was achieved by blowing low pressure air inside the membranes, which pushed the pure water in the opposite direction of filtration (from the inside out) by means of an installed air blower (compressor). For reverse washing, filtered clean water produced by the system, was used.

The cleaning process was automatic. However, both cleaning frequency and duration could be modified based on the transmembrane pressure (TMP). Washing control was achieved using a Programmable Logic Controller (PLC) and operated by automatic electric or pneumatic valves.

In addition to reverse washing, and when this method was not sufficient for solids removal from the membranes, chemical cleaning was also performed. For chemical cleaning, solutions of caustic or sodium hypochlorite and/or sulfuric acid were used to remove organic and inorganic contaminant loads respectively. Both reverse washing and chemical cleaning were performed automatically with the possibility of user intervention.

The equipment was transported to the premises of Patras’ sewage treatment plant and was installed near the secondary sedimentation tanks in an autonomous container. The installation also included a permeate water storage tank for washing cycles and a storage tank for the chemical cleaning effluents. During normal operation, the permeate overflowed its storage tank and was led through pipes for chlorination, while the concentrated stream was led to the entrance of the treatment facility and was mixed with the raw sewage to be treated. Figure 2 illustrates the installed equipment at the Patras’ treatment plant.

### 2.1. Analytical Techniques

pH and conductivity of the samples were measured by Multi 350i Set MPP 350 (WTW) while turbidity was measured with an onboard turbidity meter (one for the feed stream and one for the permeate stream). Total solids (TS), total suspended solids (TSS), chemical oxygen demand (COD), biochemical oxygen demand (BOD), and total phosphorus (TP) were determined according to “Standard methods for the examination of water and wastewater” [20]. Ion concentrations of NO_2−_-N, NO_3_-N, and Cl^−^ were measured through ionic chromatography (Dionex IC 3000), using a thermostated (30 °C) Dionex IonPac analytical column (AS19 length 4 × 250 mm, 7.5 mm I.D), a guard column (4 × 50 mm length, 12 mm I.D.), and an electron conductivity detector (Dionex). KOH solution was used as a mobile phase, under a flow rate of 0.8 mL/min with a gradient method resulting in a 3 mM KOH during equilibration and analysis, reaching 70 mM KOH during column regeneration. The total running time of analysis and the injection volume were 28 min and 10 μL respectively. NH_4_^+^-N was measured with the phenate method and total nitrogen (TN) was measured with TOC and TNM-1 unit (Shimadzu), after alkaline oxidation with persulfate method [20]. The identification of *E. coli*. and other coliforms was carried out using NPS 1035-H dishes, according to DIN EN ISO 9308-1:2014+A1:2017, and *Enterococcus* spp. with Azide-NPS 1010-H plates, according to DIN EN ISO 7899-2. Colony forming units were enumerated after serial dilutions of the fresh sample with a phosphate-buffered saline (PBS) medium and membrane filtration (0.45 μm pore-size membranes) [21]. Membranes were transferred to the hydrated medium of each dish and then were incubated at 37 °C for 21 ± 3 h for *E. coli* and 24–48 h for *Enterococci*. *E. coli*, other coliforms and *Enterococci* enumeration was performed by counting small blue, pink, and dark red colonies respectively. For the determination of sodium, potassium, iron, and manganese, samples were measured by atomic absorption spectroscopy (AAS, Perkin Elmer, AAnalyst 300).

### 2.2. Statistical Analysis

Standard deviation of the results is reported alongside their mean values and the number of independent experiments they were obtained from. The Student’s *t*-test was used to examine the statistical significance of the results obtained during process stability test and the effect of membrane filtration of permeate characteristics. Microsoft Excel was used for the statistical analysis, with a *p*-value lower than 0.05 considered significant. The results of the statistical analysis are presented in Appendix A.

## 3. Results and Discussion

After the successful installation of the membrane unit at Patras’ sewage treatment plant, the unit was tested for its ability to treat the effluent from the secondary settling tank of the plant. Part of the effluent was led to the membrane unit, prior to its chlorination. Samples were collected during different operation days and were analyzed to assess the efficiency of membrane filtration. Membrane performance and the characteristics of the membrane feed (secondary settling tank effluent), permeate, and concentrate are presented herein.

### 3.1. Membrane Performance

#### 3.1.1. Permeate Flux

The flux of membrane permeate was measured as a function of transmembrane pressure (Figure 3). In ultrafiltration membranes (UF), when convection is the prevailing mass transfer mechanism, the flux should follow a linear trend versus the applied pressure.

Figure 3 shows that the transport mechanism of the membrane process was convection, since flux was linear as a function of TMP, ranging from 6.25 to 28 L m^−2^ h^−1^ for the TMP examined (0.2–0.4 bar). This was the maximum range of TMP that could be examined using the installed equipment. It should be noted that the measured flux was stable in the timeframe examined (several hours), with no significant flux drop being observed due to the automated membrane cleaning scheme.

In a previous work of Illueca-Muñoz et al. [22], UF flux when treating the effluent of a secondary settling tank of a sewage treatment plant, was around 150 L m^−2^ h^−1^, but no data regarding the applied TMP were provided for direct comparison with the results of this study. For a similar application, Gómez et al. [23], reported a flux of 29 L m^−2^ h^−1^ for a TMP 0.1–0.6 bar, very similar to the findings of this study. On the other hand, in the work of Dialynas and Diamadopoulos [24], a permeate flux of 189 L m^−2^ h^−1^ was reported for a TMP 0.2 bar which could be attributed to the pore size of the used membrane (0.04 μm) which was larger than the pore size of the membranes installed at Patras’ treatment plant (0.01 μm). In another work examining ultrafiltration of secondary settling tank effluent by Kramer et al. [17], a tight UF (3 kDa) achieved a flux of 40 L/h/m^2^ with a TMP of 8 bar, high pressures were needed to achieve this flux because of the smaller pore size of the membrane.

#### 3.1.2. Membrane Operational Parameters

The installed equipment allowed the adjustment of the membrane operational parameters, such as permeate/feed (P/F) ratio, length of filtration cycle before membrane backwash, and duration of backwash. To examine the stability of the process, analysis of the permeate characteristics was carried out with different combinations of the aforementioned parameters, with the P/F ratios examined being 0.7, 0.75, 0.8, and 0.9. The filtration cycle lengths tested were 30, 60, and 90 min, and the backwash durations tested were 4, 6 and 8 min. The permeate COD measurements were used to assess the statistical significance of the process parameter variation. After statistical analysis of the results (Table A1), it was apparent that the effect of all three operational parameters examined had no statistically significant effect on the membrane permeate COD.

#### 3.1.3. Physicochemical Characteristics of Feed, Permeate and Concentrate Streams

Monitoring of membrane performance was carried out during a two-month period and included the analysis of the physicochemical characteristics of membrane feed, permeate, and concentrate streams. The measurements and their average values are presented in Figure 4 and Table 3 respectively. The statistical significance of the differences between feed and permeate streams was also examined (Table A2).

A drop in feed stream temperature was observed during the timespan examined due to the changing environmental conditions (Figure 4a). As was to be expected, the membrane unit had no statistically significant effect on the temperature of the permeate stream (*p*-value = 0.575). On the other hand, pH was slightly affected by the membrane (*p*-value = 0.024) (Figure 4b). Conductivity was unaffected by membrane filtration (*p*-value = 0.15), as the membrane pore size was not capable to affect ion concentration (Figure 4c). The most important improvement in the water characteristics was observed in permeate turbidity and TSS, with the membrane reducing their values by more than 99% and 94% respectively (*p*-values < 0.001) (Table 3). The membrane also exhibited a statistically significant effect on the values of TS, TP, and TN (*p*-values < 0.001) which could be attributed to the removal of TSS (Table 3). When comparing the vales of TS and TSS, it was apparent that 98% of TS were present in dissolved form. NH_4_-N also seemed to be affected (*p*-value = 0.037), but the differences were close to the statistical error. NO_2_, NO_3_, and Cl^−^ were not affected by the membrane (*p*-values of 0.255, 0.501, and 0.339 respectively). Finally, COD was significantly affected by the process (*p*-value < 0.001), with a reduction of 35%. To evaluate the COD reduction achieved by the membrane, it should be considered that most of the solids were in dissolved form.

The examined metal ions showed only small fluctuations in the values of feed, permeate, and concentrate streams due to UF’s pores which were too large to remove this type of solute.

Another impressive result was the removal of pathogens. *E. coli* and *Enterococcus* were completely removed from the secondary settling tank effluent (Table 3). Other coliforms were reduced by 95% by the membrane process. Because other coliforms were not completely absent in the membrane permeate, a chlorination test was carried out. After chlorination with 0.14% sodium hypochlorite, other coliforms were reduced to zero.

In the work of Al-Bastaki [25], during treatment of the secondary settling tank effluent through ultrafiltration, 64% reduction of COD was achieved, but not enough data were supplied for the membrane used for an accurate comparison with the results of this study. In the work of Gómez et al. [23], total removal of *E. coli* (from an initial concentration of 2.55 × 10^5^ CFU/100 mL) was reported when using UF with average pore size of 0.05 μm as tertiary treatment, while a very small number of fecal coliforms remained in the permeate of the membrane unit (from an initial concentration of 4.00 × 10^5^ CFU/100 mL). In the same study, the turbidity of the membrane effluent was less than 1 NTU. The differences from the results of this study and the results from Gómez et al. [23] may be attributed to the use of a sand filter as pretreatment to the membrane process and the characteristics of the secondary settling tank effluent that had a turbidity of 4–20, compared to the characteristics of the Patras’ treatment plant effluent that had a turbidity of 89.29 ± 15.56 NTU.

Illueca-Muñoz et al. [22] examined the use of UF (10 KDa molecular weight cutoff and microfiltration prefilter) as an option for the treatment of secondary settling tank effluent. They reported a 50% reduction of COD and total fecal coliforms removal. Once again, the turbidity reported by the researchers was less than 1 NTU. Both COD removal and turbidity of the membrane permeate were consistent with the present results.

Dialynas and Diamadopoulos [24] examined the use of UF (0.04 μm pore size) as tertiary treatment, reporting a COD reduction of 19%, permeate turbidity of less than 1 NTU, and 99.9% coliform reduction.

### 3.2. Proposed Reuse Applications

The membrane’s permeate characteristics obtained during the testing period (Table 3) were used to assess the possible water reuse applications, according to legislation (Table 1). The characteristics of the membrane feed stream (secondary settling tank effluent) were also examined, with the comparison between the two streams illustrating the benefits of implementing membrane technology.

While some of the characteristics of the secondary effluent tank adhere to the strict reuse case (c) for reuse in urban and suburban uses and the enrichment of underground aquifers used for water extraction, its turbidity and TSS (*E. coli* can be eliminated after a disinfections step) adhere only to case (a) for restricted irrigation industrial use for single use cooling water and enrichment of underground aquifer not used for drinking water extraction. As a result, overall, the effluent of Patras’ treatment plant without any further pretreatment (except for a disinfection step) could only be used for reuse case (a).

After the application of the membrane filtration, the characteristics of this effluent were improved significantly and the membrane’s permeate (Table 3) adhered to the strictest characteristics imposed by legislation (Table 1, reuse case (c)), allowing the reuse of this type of treated water for all the possible reuse cases predicted by legislation, after a disinfection step (disinfection is mandatory by the law, with 0.14% sodium hypochlorite proving capable of reducing permeate total coliform content to zero). The most promising and interesting reuse case for the reclaimed water appears to be urban and suburban use.

Urban and suburban uses include the irrigation of parks. The use of the treated sewage water for irrigation of public greeneries is a particularly attractive method for utilizing treated wastewater, especially for the proposed sewage treatment method, where the pollutant load and soluble solids were at minimum levels. The permeate water of the membrane filtration system can be safely used for irrigation of municipal parks, after being treated in the existing disinfection unit of the sewage treatment plant. A possible area to be irrigated is a municipal park near the Patras’ sewage treatment facility with a total surface of 60,000 m^2^ (Figure 5).

The installed membrane equipment is capable of producing around 300 m^3^/d of permeate. Assuming a water requirement of c.a. 2.5 cm/week [26], this amount of water would be capable of sustain 84,000 m^2^ of lawn, exceeding the requirements of the proposed irrigation area. The excess of reclaimed water, especially during colder seasons, can be used for other applications. Apart from the irrigation of public parks, urban and suburban uses include street cleaning. Every week, several street markets take place at Patras, with increased water needs for street cleaning. Another use could be waste bin washing.

## 4. Conclusions

The installed UF was tested regarding its suitability and efficiency to treat the effluent of the secondary settling tank of Patras’ treatment plant. The physicochemical characteristics of the membrane permeate stream showed significant improvements in the quality of the produced water. More specifically, TSS were reduced by more than 94%, while COD was reduced by 37%. Pathogens were detected at high concentrations in the feed stream, while *E. coli* and *Enterococcus* were absent in the membrane permeate.

The permeate flux exhibited a linear correlation to the applied TMP, indicating convection as the main mass transfer mechanism, as was to be expected for a UF membrane. Permeate to feed ratio during the operation of the membrane unit did not have a significant impact on the resulting permeate in the time frame examined in this study, even when the P/F was adjusted to 0.9.

Because of some small fluctuations in the characteristics of the secondary settling tank effluent from day to day, the membrane performance was examined for a period of about two months. Regardless of these fluctuations, the membrane was capable of consistently improving the characteristics of the treated water. Moreover, the effect of delayed intermittent cleaning was examined, with continuous operation of the membrane for 90 min not showing any deterioration of the permeate turbidity, or the onset of fouling, that would have been identified as an increase in the TMP of the process. The results presented herein indicate that the installed equipment was capable of producing improved quality water. The membrane unit operated without presenting any problems, even when pushed beyond its optimum operational conditions (in terms of P/F ratio and continuous operation without intermittent cleaning).

When comparing the characteristics of the membrane permeate with Greek legislation requirements, it becomes apparent that the quality of the produced water allows its reuse for any of the applications described in current legislation.

## Figures and Tables

**Figure 1 membranes-11-00663-f001:**
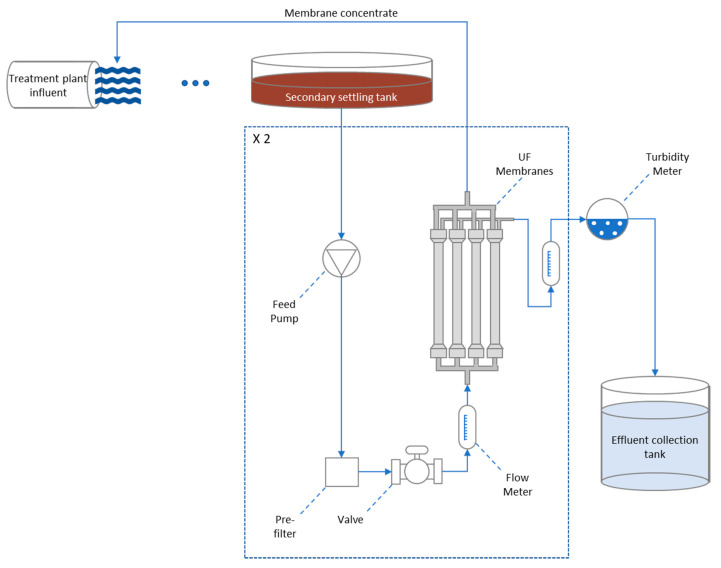
Membrane unit setup schematic.

**Figure 2 membranes-11-00663-f002:**
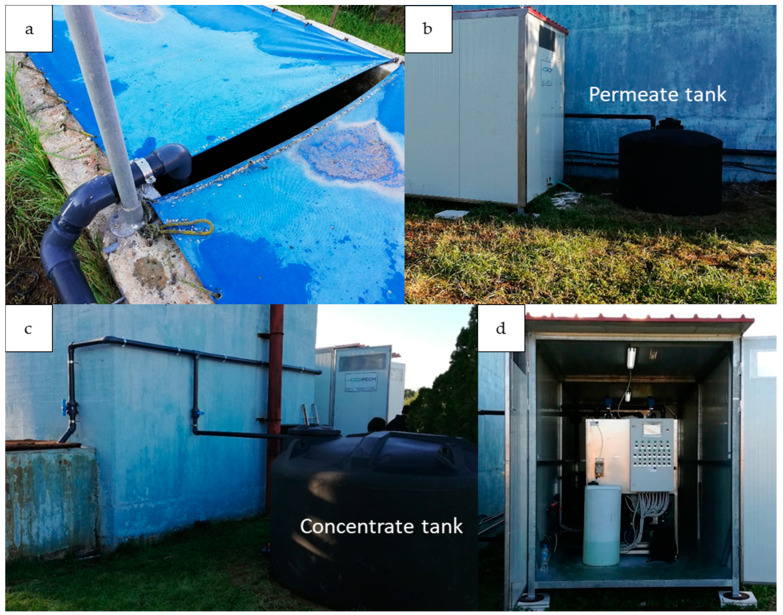
Pilot UF membrane unit installation at Patras’ wastewater treatment plant, (**a**) membrane feed, (**b**) permeate tank, (**c**) concentrate tank, and (**d**) membrane unit housing.

**Figure 3 membranes-11-00663-f003:**
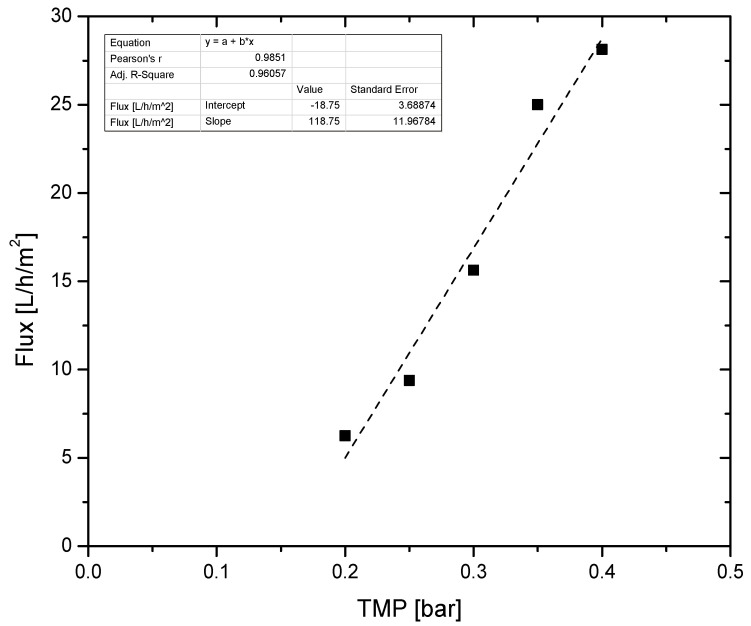
UF membrane observed permeate flux as a function of TMP.

**Figure 4 membranes-11-00663-f004:**
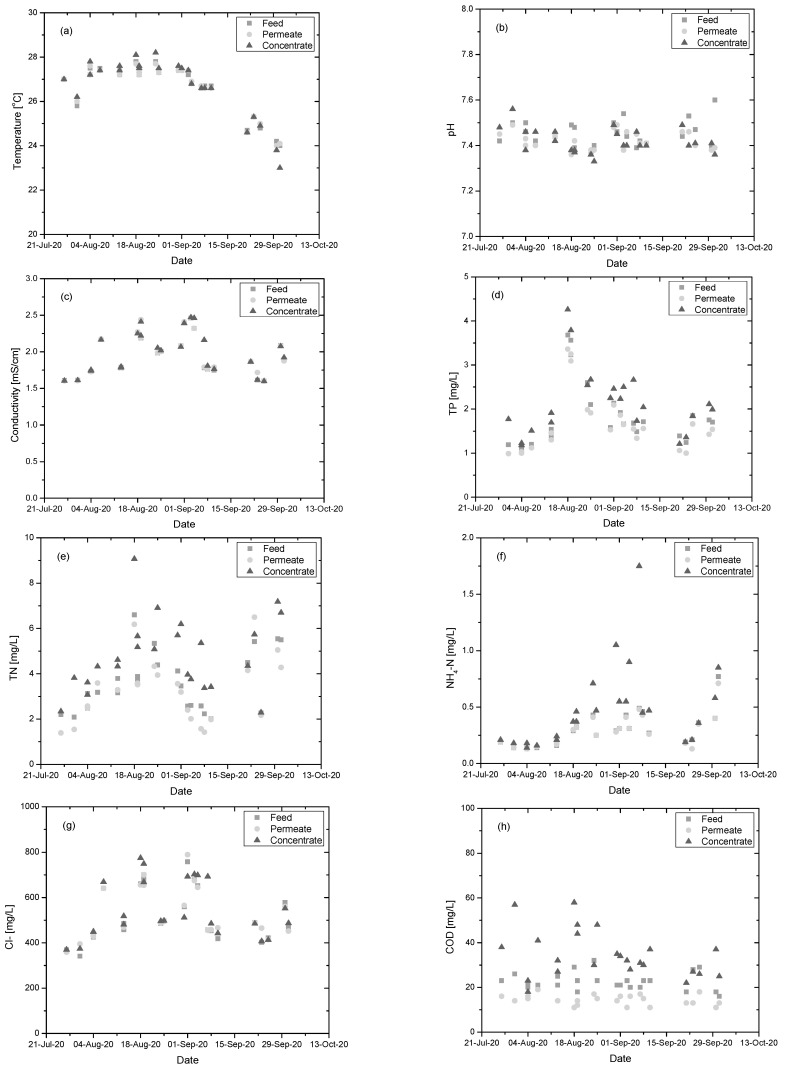
Monitoring of membrane feed, permeate, and concentrate stream characteristics: (**a**) temperature, (**b**) pH, (**c**) conductivity, (**d**) TP, (**e**) TN, (**f**) NH_4_^+^-N, (**g**) Cl, and (**h**) COD.

**Figure 5 membranes-11-00663-f005:**
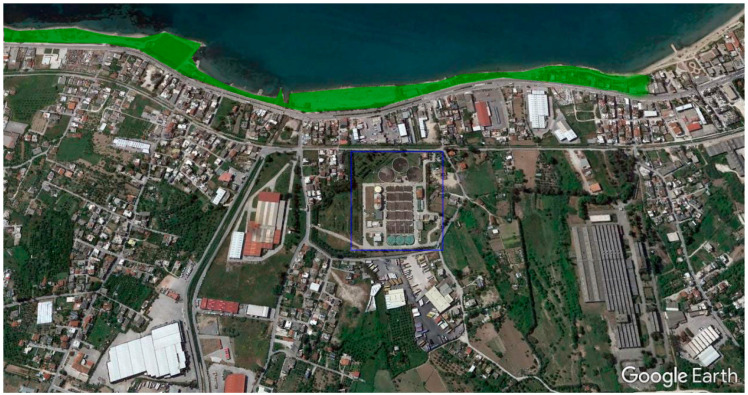
Proposed reuse application (urban use, irrigation of municipal park marked with green color) for the reclaimed water through the membrane filtration of Patras’ sewage treatment plant (marked with blue color) effluent.

**Table 1 membranes-11-00663-t001:** Greek legislation treated water quality criteria for (a) restricted irrigation, single use industrial cooling water, and enrichment of underground aquifers for nonpotable uses, (b) unrestricted irrigation and industrial use, except for single use cooling water, (c) urban and suburban reuse and the enrichment of aquifers with boreholes.

Parameter	(a)	(b)	(c)
Escherichia coli (CFU/100 mL)	<200	<5	N.A.
Total coliform bacteria (CFU/100 mL)	N.A.	N.A.	<2
BOD_5_ (mg/L)	<25	<10	<10
TSS (mg/L)	<35	<10	<2
Residual chlorine	Cont. monitoring	Cont. monitoring	Cont. monitoring
Turbidity (NTU)	N.A.	<2	<2
Total nitrogen (mg/L)	<15 *	<15 *	<15 *
Total phosphorus (mg/L)	<2 *	<2 *	<2 *
Minimum treatment required	Secondary treatment and disinfection	Secondary, tertiary treatment, and disinfection	Secondary, advanced treatment (ultrafiltration or equal treatment), and disinfection

* Only applicable if water flows to water bodies prone to eutrophication.

**Table 2 membranes-11-00663-t002:** Technical specifications of installed equipment.

Parameter	Value
Number of parallel treatment lines	2
Membrane modules	8 UF membrane modules (hollow fiber) LH3-1060
Membrane material	PVC
Filtration mode	Cross flow
Operational temperature	5–38 °C
Operational pH	2–13
Reverse flow cleaning frequency	20–60 min
Membrane pore size	0.01 μm
Membrane MWCO	50,000 Da
Permeate suspended solids	<2 mg/L
Feed capacity	400 m^3^/d
Permeate flow rate	280–360 m^3^/d
Concentrate flow rate	40–120 m^3^/d
Feeding pumps	2 × 2.2 kW

**Table 3 membranes-11-00663-t003:** Average physicochemical characteristics of feed, permeate, and concentrate streams during the timespan of membrane testing.

Parameter	Unit	Number of Experiments	Membrane Feed	Membrane Permeate	Membrane Concentrate
pH		24	7.45 ± 0.06	7.42 ± 0.04	7.42 ± 0.05
Conductivity	mS/cm	24	1.95 ± 0.28	1.96 ± 0.28	1.98 ± 0.29
Turbidity	NTU	24	89.29 ± 15.56	<1	31.15 ± 11.04
TSS	mg/L	24	16.45 ± 7.19	<1	26.68 ± 1.38
TS	mg/L	17	831 ± 159	597 ± 132	979 ± 198
TP	mg/L	23	1.86 ± 0.74	1.69 ± 0.69	2.21 ± 0.83
TN	mg/L	24	3.62 ± 1.32	3.25 ± 1.4	4.83 ± 1.63
NH_4_^+^-N	mg/L	24	0.30 ± 0.15	0.29 ± 0.14	0.48 ± 0.37
NO_2_^−^-N	mg/L	24	0.13 ± 0.07	0.13 ± 0.07	0.17 ± 0.07
NO_3_^−^-N	mg/L	24	1.18 ± 0.88	1.15 ± 0.93	1.47 ± 0.97
Cl^−^	mg/L	24	521 ± 118	526 ± 113	545 ± 126
COD	mg/L	24	22.63 ± 3.93	14.38 ± 2.26	34.50 ± 10.48
BOD	mg/L	9	6.67 ± 3.2	4.78 ± 3.19	8.45 ± 4.37
Na	mg/L	1	86.70	92.80	94.25
K	mg/L	1	16.15	19.66	18.35
Fe	μg/L	1	87.00	63.00	107.00
Mn	μg/L	1	63.00	56.00	67.00
*E. coli*	CFU/100 mL	4	(6.12 ± 7.78) × 10^4^	N.D.	(1.67 ± 1.59) × 10^5^
*Enterococcus*	CFU/100 mL	4	(1.87 ± 2.85) × 10^5^	N.D.	(5.27 ± 4.48) × 10^5^
Other Coliforms	CFU/100 mL	4	(5.05 ± 4.47) × 10^4^	(5.35 ± 2.6) × 10^3^	(8.60 ± 3.68) × 10^3^

## Data Availability

Data are contained within the article in Section 3: Results.

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
