# Peer review of "Recovery of Water from Secondary Effluent through Pilot Scale Ultrafiltration Membranes: Implementation at Patras’ Wastewater Treatment Plant"

_membranes, 2021, doi:10.3390/membranes11090663_

Round 1

Reviewer 1 Report

The manuscript deals with the recovery of water from secondary effluent through the use ofultrafiltration membranes and its implementation at Patras’ wastewater treatment plant.

The general work appear interesting and useful. Aims of the work are clearly highlighted in the Introduction section. Experimental results related to the membrane performance (permeate flux, membrane operational parameters and physico-chemical characteristics of feed, permeate and retentate streams) are well presented and discussed.

Based on the above comments the current version of the manuscript can be accepted for publication.

Author Response

Reviewer 1

The manuscript deals with the recovery of water from secondary effluent through the use of ultrafiltration membranes and its implementation at Patras’ wastewater treatment plant.

The general work appear interesting and useful. Aims of the work are clearly highlighted in the Introduction section. Experimental results related to the membrane performance (permeate flux, membrane operational parameters and physico-chemical characteristics of feed, permeate and retentate streams) are well presented and discussed.

Based on the above comments the current version of the manuscript can be accepted for publication.

The authors would like to thank the reviewer the comments.

Reviewer 2 Report

Authors presented results on the application of the commercial UF modules for the recovery of water from secondary effluent. The data were collected at real wastewater treatment plant. The manuscript fits very well the profile of Membranes journal. The text is very well organized, information is clearly presented. Authors should only improve Fig. 2 because is illegible. Moreover, instead of "permeate" or "concentrate" only tanks can be seen. 
Another issue - could Authors compare the results of their investigations with results of other Authors?

Final recommendation - minor revision. 

Author Response

Reviewer 2

Authors presented results on the application of the commercial UF modules for the recovery of water from secondary effluent. The data were collected at real wastewater treatment plant. The manuscript fits very well the profile of Membranes journal. The text is very well organized, information is clearly presented. Authors should only improve Fig. 2 because is illegible. Moreover, instead of "permeate" or "concentrate" only tanks can be seen. 

Figure 2 has been modified according to the reviewer’s comments, it is now clarified in the caption that the figure presents the permeate and concentrate tanks, which are also indicated with labels in the figure. Regarding the quality of the figure, the authors have provided high quality pictures, but maybe they have been downgraded from the reviewing process.

Another issue - could Authors compare the results of their investigations with results of other Authors?

Comparison of the results of this work with literature data is presented in Lines 238-249 and Lines 294-314.

 We would like to thank the reviewers for their time and comments that allowed the authors to improve the manuscript